# Community Detection with the Triplet Loss

**Marc Lelarge**
INRIA
École normale supérieure, CNRS, PSL Research University
75005 Paris, France
`marc.lelarge@inria.fr`

## Abstract

We present a scalable approach for unsupervised learning on graph-structured data based on a simple graph embedding learned via the triplet loss. For the community detection problem on the stochastic block model, our algorithm is optimal, with the same performance as the spectral technique based on the Bethe Hessian. On synthetic low-dimensional datasets, our algorithm generalizes well, having state of the art performances. In a semi-supervised learning framework, our algorithm extends naturally and incorporates the additional information with a great increase in performances.

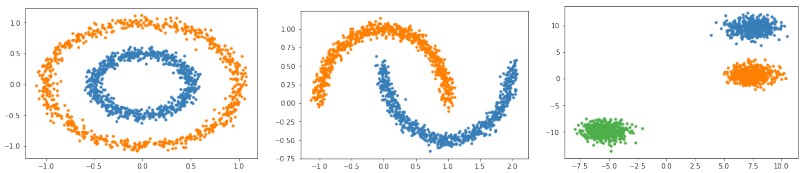

Figure 1: Illustrative 2D examples showing the result of our clustering algorithm to be compared with scikit-learn clustering algorithms `http://scikit-learn.org/stable/modules/clustering.html`

## 1 Introduction

Clustering and community detection is a fundamental unsupervised data analysis task. Efficient algorithms work by first embedding the data in a low dimensional space and then applying a standard clustering algorithm (like $k$-means) on this representation to obtain the clusters. Spectral clustering (Von Luxburg (2007)) is a leading and highly popular clustering algorithm. It constructs the embedding from the leading eigenvectors of appropriate operators defined on the similarity graph. However, to ensure good performances, the operator needs to be carefully chosen depending on the characteristics of the similarity graph. In this paper, we develop a generic approach and propose a simple algorithm to learn an embedding of a graph by using the triplet loss (Schroff et al. (2015)). Each edge of the graph is interpreted as a suggestion to put both endpoints in the same community or cluster and the triplet loss will leverage this observation by discriminating edges from non-edges. Note that a Siamese net architecture has been recently proposed by (Shaham et al. (2018)) in order to learn similarities between data points which is then used in learning the spectral embedding. Here, we use a contrastive loss to learn directly the embedding. A triplet network was used in (Hoffer & Ailon (2015)) to learn metric in a supervised setting (sampling of positive and negative pairs is done thanks to the labels). In our case, we only use the graph structure as a noisy pairing of positive pairs (most edges connect positive pairs but there is a positive fraction of errors). Our algorithm can be seen as a denoiser to recover positive pairs, i.e. the structure of clusters hidden in the graph.

Our algorithm is highly scalable as it takes as input a stream of incoming edges, it only requires a constant memory per node to store the low dimensional representation of each node. For each incoming edge, our algorithm needs to sample a node at random in the network. We use a muticlass hinge loss and SGD to learn the embedding of the nodes. A standard clustering algorithm is then

used directly on these embeddings. Figure 1 illustrates results obtained by our algorithm on some 2D examples. In the rest of the paper, we will concentrate on the stochastic block model (SBM), a popular generative model for random graphs with community structure. We will consider very sparse SBMs for which recent spectral algorithms have been designed and shown to be optimal (Saade et al. (2014)). Sparse graphs are much more difficult to cluster but very relevant from a practical perspective, in particular when dealing with very large datasets. Indeed, as shown in (Saade et al. (2016)), with our clustering algorithm on sparse graphs, we need to compute a number of pairwise similarities scaling only linearly with the number of data points.

## 2 DESCRIPTION OF THE ALGORITHM

In order to motivate our algorithm, we present a standard semidefinite relaxation of our partition problem. Given a (undirected) graph $G = (V, E)$ with $n = |V|$ vertices, we denote by $A$ its (symmetric) adjacency matrix, i.e. $A_{ij} = 1$ whenever $(i, j) \in E$. The minimum bisection problem consists in finding a cut of the set of vertices $V$ in two sets of equal size as to minimize the number of edges across the partition. Formally:

$$\boldsymbol{\sigma}^* = \arg\max_{\boldsymbol{\sigma} \in \{\pm 1\}^n} \left\{ \sum_{(i,j) \in E} \sigma_i \sigma_j : \sum_i \sigma_i = 0 \right\}.$$

This problem is hard to approximate (Khot (2006)) which motivates the following SDP relaxation:

$$\max_{\underline{\boldsymbol{\sigma}}} \sum_{(i,j) \in E} \langle \boldsymbol{\sigma}_i, \boldsymbol{\sigma}_j \rangle, \tag{1}$$

where $\underline{\boldsymbol{\sigma}} = (\boldsymbol{\sigma}_1, \ldots, \boldsymbol{\sigma}_n)$ and $\boldsymbol{\sigma}_i \in \mathbf{R}^n$ is the vector associated to node $i$ and must satisfy the constraints $\|\boldsymbol{\sigma}_i\|_2 = 1$ and $\sum_{i=1}^n \boldsymbol{\sigma}_i = 0$ for all $i \in [n]$. In order to obtain a low embedding of the graph, we will consider the case where each $\boldsymbol{\sigma}_i \in \mathbf{R}^m$ with $m < n$, indeed we fixed $m = 32$ in our experiments.

At each time, our algorithm stores a vector $\boldsymbol{\sigma}_i$ for each node $i \in V$. Consider a setting, where edges arrive in a stream in random order and suppose edge $(i, j) \in E$ arrives. Then our algorithm increments its current loss by:

$$\ell_{(i,j)} = \max\left( \langle \boldsymbol{\sigma}_i, \boldsymbol{\sigma}_k \rangle - \langle \boldsymbol{\sigma}_i, \boldsymbol{\sigma}_j \rangle + \alpha; 0 \right), \tag{2}$$

where $k$ is a node chosen uniformly at random in $V$ and $\alpha$ is a fixed parameter. Typically, in a sparse graph, we will have $(i, k) \notin E$. Also, if there are 2 communities of the same size, then $i$ and $k$ are in the same community with probability $1/2$ and in different communities with the same probability. By symmetry, we expect that on average $\langle \boldsymbol{\sigma}_i, \boldsymbol{\sigma}_k \rangle \approx 0$. Now, note that the loss $\ell_{(i,j)}$ is zero as soon as $\langle \boldsymbol{\sigma}_i, \boldsymbol{\sigma}_k \rangle + \alpha \le \langle \boldsymbol{\sigma}_i, \boldsymbol{\sigma}_j \rangle$. Once our algorithm received a batch of edges, it will update all the embeddings $\boldsymbol{\sigma}_i$ in order to minimize its loss using backpropagation. Replacing $\langle \boldsymbol{\sigma}_i, \boldsymbol{\sigma}_k \rangle \approx 0$ in the expression above, we see that our algorithm will try to enforce $\langle \boldsymbol{\sigma}_i, \boldsymbol{\sigma}_j \rangle \ge \alpha$ for all $(i, j) \in E$. We want to keep the variations of the random $\langle \boldsymbol{\sigma}_i, \boldsymbol{\sigma}_k \rangle$ of the order of the dimension $m$ of the embedding, so that we chose $\alpha \approx m$. This will ensure that our algorithm almost maximizes (1).

**Practical considerations:** the graph is encoded as a list of edges. Note that the loss (2) is not symmetric in $i$ and $j$, hence each edge will appear twice in the list, as the ordered pairs $(i, j)$ and $(j, i)$. We used stochastic gradient descent with batch sizes of 256 edges. We also used a L2 regularization. One epoch corresponds to two passes on each edge (one in each direction). After $k$ epochs, the embedding of a node will incorporate the information from nodes at graph-distance at most $k$ from it. In order to get good performances, the number of epochs needs to scale with the diameter of the graph to ensure propagation of the information over the graph. Given an embedding $\boldsymbol{\sigma}_i$ of each node, the partition can be done by a standard spectral clustering in $\mathbb{R}^m$. In our case, we use the singular-value decomposition of the matrix $\underline{\boldsymbol{\sigma}} = (\boldsymbol{\sigma}_1, \ldots, \boldsymbol{\sigma}_n) \in \mathbb{R}^{m \times n}$ and then clusters are assigned according to the sign of the components of the first right singular vector. In order to improve the performance of the algorithm, the size of the cut resulting from such a clustering can be computed every few epochs, if the cut size increases, we can roll-back to the more efficient embedding.

**Semi-supervised setting:** when some nodes are known to belong to cluster one or two, we modify the algorithm by simply fixing their embedding to the (properly normalized) vectors $(\underbrace{+1, \ldots, +1}_{n/2}, \underbrace{-1, \ldots, -1}_{n/2})$ and $(\underbrace{-1, \ldots, -1}_{n/2}, \underbrace{+1, \ldots, +1}_{n/2})$ respectively.

## 3 RESULTS

We now give some theoretical justifications in the case of the SBM with two communities (for the sake of simplicity), i.e. a random graph generated as follows: we partition $V = V_+ \cup V_-$ in two equal size communities (this is the planted partition); conditional on the partition, edges are independently drawn between each pair of nodes $(i, j)$ with probability $a/n$ if $i$ and $j$ belong to the same community and with probability $b/n$ if they are in different community. This model has a long history and results in (Decelle et al. (2011)) have triggered a renewed interest in the regime where $n \to \infty$ while $a$ and $b$ are kept fixed. The signal to noise ratio turned out to be $SNR = \frac{(a-b)^2}{2(a+b)}$, in the sense that if $SNR < 1$ then no signal can be detected in the graph, while as soon as $SNR > 1$ then a partition with a positive overlap with the planted partition can be found (in polynomial time (Massoulié (2014))). If $\hat{V}_+$ and $\hat{V}_-$ is a partition of $V$, then its overlap is defined as: $overlap = 2\frac{|\hat{V}_+ \cap V_+| + |\hat{V}_- \cap V_-|}{n} - 1$. The overlap measures the quality of the reconstruction of the planted partition from zero for a pure random guess to one for exact recovery.

It turns out that our algorithm returns a partition with a positive overlap as soon as it is possible. To be more precise, previous results are theoretical results valid in the limit $n \to \infty$ and based on this asymptotic analysis, (Saade et al. (2014)) proposed a spectral embedding based on the so-called Bethe Hessian operator given by $BH(r) = (r^2 - 1)ID - rA + D$ where $r$ is the average degree in the graph. They showed that its performance is optimal in the sense that: in the large $n$ limit, the overlap is the same as the one achieved by a Bayes optimal estimator. More rigorous results (in particular in the unbalanced case) are provided in (Lelarge & Miolane (2016)).

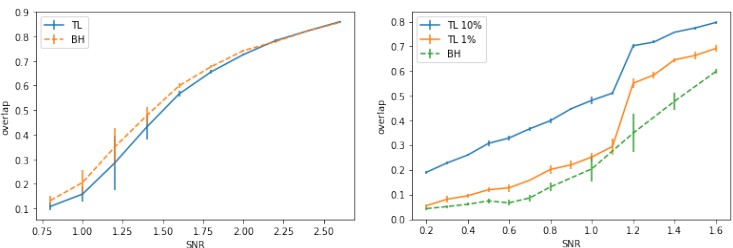

Figure 2: Left: comparison of our algorithm based on the triplet loss (TL) with the Bethe Hessian (BH). Right: semi-supervised version of our algorithm with $1\%$ and $10\%$ of labeled data compared to Bethe Hessian (BH). Note that the ranges of SNRs are not the same in both figures. The average degree in the graph is fixed to $4$ and its size is $n = 1000$. Each point corresponds to an average over 20 graphs.

Figure 2 (left) shows that our algorithm is almost optimal for SBMs in the whole range of SNR as it almost matches the performance of the spectral algorithm based on the Bethe Hessian. Note that the SDP (1) with an additional constraint on $\boldsymbol{\sigma}$ being of rank one is the maximum likelihood estimator in the case of the SBM. Robustness to perturbations of the SDP relaxation even with $m < n$ was already observed in (Javanmard et al. (2016)). Our results show that it is actually possible to obtain state of the art results in streaming, with SGD and the triplet loss.

Figure 2 (right) shows the performances of our algorithm in a semi-supervised setting with respectively $1\%$ and $10\%$ of labeled data (corresponding to an initial overlap of $0.04$ and $0.2$ resp.). The performance of the spectral algorithm based on the Bethe Hessian is given as a point of comparison. To the best of our knowledge, there is no solution to integrate the labeled data into a spectral algorithm. At $SNR = 1.6$, with only on average 4 similarities per data point and with $20\%$ of these similarities being erroneous, our algorithm achieves $80\%$ (resp. $85\%$ and $90\%$) of true positives with $0\%$ (resp. $1\%$ and $10\%$) of labeled data.

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
