# OpenReview forum: "Community Detection with the Triplet Loss"
_ICLR.cc/2018/Workshop — Reject_

### Official Review · AnonReviewer2 · 2018-03-08
**Preliminary but intriguing results on graph clustering with learned similarities**

**Rating:** 7
**Confidence:** 3

**Review:**

This short paper reports on results obtained for clustering where the similarity function is learned using minimization of the triple loss. It is reported that on very sparse networks generated by the stochastic block model that this way performance of standard spectral clustering matches the performance of spectral algorithms fine-tuned to the data coming from the stochastic block model. The paper seems to provide a good seed to understand further how data pre-processing improves performance of clustering algorithms.

---

### Official Review · AnonReviewer3 · 2018-03-09
**Promising but presentation is unclear**

**Rating:** 5
**Confidence:** 4

**Review:**


The author proposed an algorithm that performs graph clustering based on an input stream of edges. Empirically, for the 2-cluster case, across a wide range of signal-to-noise ratios, the algorithm is shown to achieve comparable performance in terms of clustering accuracy with respect to an existing algorithm that comes with theoretical guarantee, with the added benefit that it can easily incorporate pre-labeled pairs.

While the results presented are encouraging, the description of the algorithm itself is unclear to me. In particular, how is each sigma_i initialized and updated? I assume it is perhaps random initialization followed by SGD updates but what about the constraints? It is also not clear to me how the algorithm works asymptotically, e.g. with respect to (1). Also, it seems that for the labeled pairs (semi-supervised setting), the suggested embedding implies working with a full n-by-n matrix instead of m-by-n matrix. Computationally and storage-wise, it is not clear how the algorithm compares to other
spectral clustering approaches.

I believe that the author has a promising algorithm but I am afraid I cannot recommend acceptance in its current state, since it is not clear enough in the sense that an interested reader could implement and test it.

---

> ### Public Comment · ~marc_lelarge1 · 2018-04-03
> **clarification**
>
> Thank you very much for your feedback. It will help us in the revision of the paper.
> As you guessed, I took random initialization followed by SGD and actually did not encode the constraints of the SDP relaxation. it turns out the algorithm still output more or less balanced partition in the symmetric case and adapt 'naturally' in non-symmetric cases.
>
> As you noted, there is a typo in the semi-supervised setting, the planted embeddings are of sizes m and not n as written in the paper, divided in two equal parts with +1 or -1.

---

### Decision · Program_Chairs · 2018-03-20
**ICLR 2018 Workshop Acceptance Decision**

**Decision:**

Reject

**Comment:**

Based on the reviews, this paper has not been accepted for presentation at the ICLR workshop. However, the conversation and updates can continue to appear here on OpenReview.